# Phylogenetic analysis of the distribution of deadly amatoxins among the little brown mushrooms of the genus *Galerina*

**Brandon Landry**[1], **Jeannette Whitton**[1], **Anna L. Bazzicalupo**[1], **Oldriska Ceska**[2], **Mary L. Berbee**[1] *

**1** Department of Botany, University of British Columbia, Vancouver, British Columbia, Canada, **2** 1809 Penshurst Road, Victoria, British Columbia, Canada

* mary.berbee@gmail.com

**Data Availability Statement:** All sequences are available through GenBank and accession numbers are listed in Supplementary S1 Fig. The alignment is available through DRYAD https://doi.org/10.5061/dryad.r7sqv9s9z.

## Abstract

Some but not all of the species of 'little brown mushrooms' in the genus *Galerina* contain deadly amatoxins at concentrations equaling those in the death cap, *Amanita phalloides*. However, *Galerina*'s ~300 species are notoriously difficult to identify by morphology, and the identity of toxin-containing specimens has not been verified with DNA barcode sequencing. This left open the question of which *Galerina* species contain toxins and which do not. We selected specimens for toxin analysis using a preliminary phylogeny of the fungal DNA barcode region, the ribosomal internal transcribed spacer (ITS) region. Using liquid chromatography/mass spectrometry, we analyzed amatoxins from 70 samples of *Galerina* and close relatives, collected in western British Columbia, Canada. To put the presence of toxins into a phylogenetic context, we included the 70 samples in maximum likelihood analyses of 438 taxa, using ITS, RNA polymerase II second largest subunit gene (*RPB2*), and nuclear large subunit ribosomal RNA (LSU) gene sequences. We sequenced barcode DNA from types where possible to aid with applications of names. We detected amatoxins only in the 24 samples of the *G. marginata* s.l. complex in the *Naucoriopsis* clade. We delimited 56 putative *Galerina* species using Automatic Barcode Gap Detection software. Phylogenetic analysis showed moderate to strong support for *Galerina* infrageneric clades *Naucoriopsis*, *Galerina*, *Tubariopsis*, and *Sideroides*. *Mycenopsis* appeared paraphyletic and included *Gymnopilus*. Amatoxins were not detected in 46 samples from *Galerina* clades outside of *Naucoriopsis* or from outgroups. Our data show significant quantities of toxin in all mushrooms tested from the *G. marginata* s.l. complex. DNA barcoding revealed consistent accuracy in morphology-based identification of specimens to *G. marginata* s.l. complex. Prompt and careful morphological identification of ingested *G. marginata* s.l. has the potential to improve patient outcomes by leading to fast and appropriate treatment.

## Introduction

*Galerina*, a genus of small, yellow-orange or yellow-brown mushrooms, includes species that have been implicated in dozens of poisoning cases worldwide [1]. However, information

**Funding:** This research was funded by grants from the Natural Sciences and Engineering Research Council of Canada (https://www.nserc-crsng.gc.ca/index_eng.asp) including a Canada Graduate Scholarship-Master's Program and a Michael Smith Foreign Travel Supplement to BL, and a Discovery Grant RGPIN-2016–03746 to MLB.

**Competing interests:** The authors have declared that no competing interests exist.

about exactly which of the >300 species in the genus [2] pose a poisoning risk is incomplete and confusing. This is partly because *Galerina* species are difficult to identify using just morphological characters. In part, toxin analysis has usually involved destructive sampling, leaving no voucher material to confirm identification. DNA barcoding has not previously been applied to link identifications of specimens with toxin analysis, and toxins have not been assayed from diverse *Galerina* species. Here, we connect vouchered *Galerina* specimens to DNA barcode sequences and to amatoxin presence and absence in the context of the most complete molecular phylogeny of the genus to date.

Although individual *Galerina* mushrooms are small, the amatoxins can have dramatic consequences if ingested. Given the amatoxin $LD_{50}$ (amount of substance required to kill 50% of the test population) of 0.1 mg/kg body weight, 10 fruiting bodies of one of the toxic species would be sufficient to poison a child weighing 20 kg [1]. Serious illness has resulted in people of various ages when *Galerina* mushrooms have been confused with edible or hallucinogenic mushrooms and eaten in quantity. By the time serious symptoms appear, 2–4 days after eating mushrooms, the toxins have inflicted serious damage on the liver and other internal organs. A family in Japan including a six-year-old boy ate soup containing what were probably *Galerina fasciculata*, possibly mistaken for wild enoki mushrooms [3]. The older family members experienced nausea and diarrhea and then recovered, but the boy's condition became progressively worse. Some 36 hours after eating the soup, the boy was admitted to the hospital; 72 hours after the meal, his liver failed. Following treatment, he slowly recovered, to be discharged after 15 days [3]. A 32-year-old Swedish woman sautéed and ate *Galerina marginata*, mistaking them for honey mushrooms (*Armillaria* species). She was admitted to the hospital 17 h later with vomiting and diarrhea, and with blood enzyme levels indicating liver damage [4]. She recovered after nine days in the hospital. Two days after their cafeteria erred by serving a locally sourced 'mushroom dish' that likely contained *Galerina sulcipes*, a group of 13 coworkers in China, aged 19–56 required 10 days of hospitalization to recover from liver and kidney damage [5]. Although details are unavailable, in 2011, three *Galerina* poisoning cases including one fatality were reported in North America [6]. There is no known antidote for amatoxin ingestion, but case studies show that supportive therapy, such as replacing electrolytes and keeping the patient hydrated saves lives [7, 8]. Better knowledge of the taxonomic distribution of amatoxin production may allow for better documentation of the geographic range and abundance of toxic species. If ingested mushrooms can be identified as amatoxin-containing species earlier, appropriate treatment can be initiated earlier, likely improving outcomes.

Deadly amatoxins in *Galerina* mushrooms have been documented since the mid-20th century. In 1954, two patients consumed what was later identified as *Galerina venenata* and presented with symptoms mirroring poisoning by the death cap, *Amanita phalloides* [9]. Prompted by these poisoning cases, Tyler and Smith [10] used paper chromatography to show that *G. venenata* contains α- and β-amanitin–two of the amatoxins, the toxic peptides identified from the genus *Amanita*.

To discuss the relationships of the toxin producers among the large number of *Galerina* species, infrageneric clades become relevant. A series of authors have subdivided the genus into subgenera and sections; e.g. Gulden and Hallgrímsson [11] and Smith and Singer [12]. The infrageneric taxa applied by different authors are only partially congruent with one another or with molecular phylogenies [13]. For clarity of communication, Gulden et al. [13] designated four infrageneric clades in their molecular phylogenies as informal groups "*Naucoriopsis*," "*Galerina*," "*Tubariopsis*," and "*Mycenopsis*," pointing out that the names "largely reflect already recognized morphology-based subgenera or sections within *Galerina*." Our results are largely congruent with these earlier studies and so we recognize Gulden et al. [13]'s four

provisional clades as subgenera. We also apply "*Sideroides*" as a subgenus, based on an infrageneric taxon first used in Smith and Singer's monograph [12].

Previous phylogenetic and toxin studies placed known *Galerina* toxin-producers in subgenus *Naucoriopsis* [1, 13]. Within *Naucoriopsis*, amatoxins have been reported in the *G. marginata* s.l. species complex [1]. Five other species that are also reported to contain amatoxins are likely to be members of *Naucoriopsis*, although without verification by DNA barcoding. Muraoka et al. [14] and Muraoka and Shinozawa [15] purified amatoxins from cultures of *G. fasciculata* and *G. helvoliceps*. Besl [16] extracted amatoxins from cultures of *G. beinrothii*; from dried mushrooms of *G. badipes*; and from both cultures and dried mushrooms of the *G. marginata* species complex. Besl et al. [16] also reported negative results; toxins were not detected from four specimens selected from among the ~200 *Galerina* species from outside *Naucoriopsis*.

Of the toxin producers associated with specimen vouchers, the culture *Galerina* 'marginata' CBS 339.88 is the best studied. The Joint Genome Institute sequenced its complete genome. Luo et al. [17] characterized its genes responsible for α-amanitin synthesis and used hybridization to indicate that the same genes are present in *G. venenata* CBS 924.72, and *G. badipes* (CBS 268.50). Surprisingly, *G. badipes* reportedly produced γ-amanitin but not the more common α- or β-amanitin [16].

The number of *Galerina* species that produce toxins is unclear. Until recently, most *Galerina* species have been described and delimited based on micro- and macromorphological differences. Smith and Singer's [12] monograph on the genus distinguished 199 species of *Galerina*. However, Gulden et al. [18] showed that nuclear ribosomal internal transcribed spacer (ITS) sequence variation did not support the monophyly of species from vouchers labeled *G. marginata*, *G. autumnalis*, *G. unicolor*, *G. oregonensis*, and *G. venenata*. Gulden et al. synonymized all of these under *G. marginata*. The study left unclear whether other species should be included in *G. marginata*. The possibility remained that cryptic species may be contained in a group that we refer to as '*G. marginata* s.l.'.

*Galerina* appears polyphyletic in molecular phylogenies that draw on ITS and large ribosomal subunit (LSU) data [13, 18]. *Gymnopilus* appears nested within *Galerina*'s subgenus *Mycenopsis* with a Bayesian posterior probability of 1.00. Other *Galerina* species were intermingled with *Phaeocollybia*, *Hebeloma* and other genera, mostly without strong Bayesian support [13]. Suggesting that some of the apparent *Galerina* polyphyly reflected lack of data, when Matheny et al. [19] used more data, 4508 aligned sites from a combination of ribosomal and *RPB2* (encoding the RNA polymerase II second largest subunit B150) gene sequences, phylogenies no longer showed *Phaeocollybia* and *Hebeloma* intermingled with *Galerina*. Matheny et al. transferred *Galerina clavus*, which was clearly not a *Galerina*, to a new genus, *Romagnesiella* [19]. These results suggested encouragingly that including *RPB2* with ribosomal gene data might clarify the infrageneric structure of *Galerina*, putting the toxic species in a larger phylogenetic context.

Our goal was to resolve relationships and clarify the phylogenetic distribution of amatoxins among *Galerina* species. To more closely characterize poisonous species, we aimed to analyze DNA and toxins of vouchered *Galerina* collections from the UBC Herbarium in the Beaty Biodiversity Museum (https://herbweb.botany.ubc.ca/herbarium/search.php?Database=fungi). Many of these are recently accessioned collections made by regional mycologists, especially Oldriska Ceska and Paul Kroeger. Discovering which *Galerina* species contain amatoxins is technically straightforward because a small amount of fungal tissue suffices for both toxin analysis and DNA barcoding. Two studies [20, 21] have demonstrated that amatoxins are readily detected and quantified via liquid chromatography-mass spectrophotometry from as little as 8 mg dried *Amanita*, even in herbarium specimens that were 17 years old. Using preliminary

ITS phylogenies to represent the diversity of clades in *Galerina*, we selected specimens for toxin analysis and for sequencing of partial LSU and *RPB2* regions. To help guide applications of names, we borrowed specimens including types determined by A.H. Smith and sequenced their ITS1 regions. We quantified α-amanitin concentrations from a diverse sample of 62 DNA-barcoded UBC *Galerina* specimens and eight species from closely related genera. Integrating toxin data in a broad phylogenetic framework gives us new power to predict toxicity from morphology and to speed identifications of specimens involved in possible poisoning cases.

## Materials and methods

### Taxon sampling, DNA amplification and phylogenetic analysis

For this study, we re-analyzed ITS sequences of *Galerina* specimens from UBC determined previously by Bazzicalupo et al. [22]. For each collection, DNA extraction, PCR amplification, and ITS sequencing had been replicated [22]. We analyzed the ITS sequences of 147 *Galerina* collections from which we recovered the same sequence in each of two independent extractions (S1 Table). For *RPB2* and LSU amplifications, we extracted additional DNA from specimens selected to represent the diversity of lineages as estimated from preliminary analyses of the ITS data. We extracted DNA from 5–20 mg of gill tissue following instructions in the Qiagen DNEasy Plant Mini Kit for PCR amplification with Illustra PuReTaq Ready-To-Go PCR beads (GE Healthcare: Mississauga, ON, Canada). We used primers LR0R and LR5 [23] for LSU gene amplifications. For *RPB2*, we initially used primers bRPB2-6F and bRPB2-7.1R [24]. The PCR cycles began with an initial denaturation at 95˚C for 5 min, followed by 30 cycles of 95˚C denaturation for 30 sec, 55˚C annealing for 30 sec, 72˚C elongation for 30 sec, increasing the elongation time by 4 sec per cycle and concluding with a final elongation at 72˚C for 7 minutes. For *RPB2* samples that gave only weak bands or no bands at all, we re-amplified the product in nested PCR reactions using bRPB2-7R [24] and a re-designed internal forward primer berniF 5' ATG GTG TGC CCT GCG GAA AC. For forward and reverse Sanger sequencing, we used BigDye Terminator v3.1 (Thermo Fisher Scientific: MA, USA) following the manufacturer's instructions. The UBC Sequencing and Bioinformatics Consortium performed the electrophoresis.

The 368 ITS sequences analyzed included 161 sequences from UBC specimens of *Galerina*, *Hebeloma* and *Gymnopilus*, genera representing the family Hymenogastraceae. To help associate names with clades, we sequenced the ITS1 regions from 14 *Galerina* specimens from MICH and examined by A.H. Smith, including types where possible. Also to help associate names with clades, we used sequences from Gulden et al. [13, 18]. We used a series of BLAST searches to select additional GenBank sequences to represent the known diversity in the genus and we included ITS sequences of *Psilocybe* in addition to *Galerina*, *Hebeloma* and *Gymnopilus* in the analysis. We selected 154 sequences from the 5' end of the LSU, 28 of them determined for this study, and 78 *RPB2* sequences, 24 from this study to represent *Galerina* and closely-related families Hymenogastraceae, Strophariaceae, Crepidotaceae, Inocybaceae, Tubariaceae, Bolbitiaceae and Cortinariaceae. For voucher information and GenBank accession numbers, see S1 Table.

We used the MAFFT online server with the L-INS setting [25] to obtain initial alignments for each locus, then refined the alignments manually using Mesquite 3.5 [26]. For the *RPB2* dataset, we excluded introns from the final alignment. Using jModelTest 2 [27] implemented on the CIPRES portal [28], we selected, as best nucleotide substitution models, (AICc) GTR+I +G for the ITS and LSU datasets; TIM1+I+G for *RPB2* codon position 1; and TVM+I+G for *RPB2* codon positions 2 and 3. For analyses, we approximated the best models using GTR+I

+G throughout. For each individual alignment and the concatenated alignment, we used RAxML v.8.2.10 [29] on the CIPRES portal to infer maximum likelihood trees from 200 'thorough' searches. We used 500 bootstrap replicates to assess branch support. Conflicts in the topologies from individual loci generally involved weakly to moderately supported nodes (<70% bootstrap) (S2–S4 Figs), so we concatenated the alignments in Mesquite.

For subsequent analyses of concatenated LSU, *RPB2* and ITS data, the ITS regions of the more distant outgroups were too variable to align, and so we included only species of *Galerina*, *Gymnopilus*, *Psilocybe* and *Hebeloma*. We included sequence data from each specimen analyzed for toxins and from each specimen represented by data from LSU or *RPB2* sequence regions. We included a representative of each unique ITS haplotype. To increase geographical sampling, we included a representative of each country of origin from among sequences with the same haplotype. The resulting dataset included 337 taxa and 4401 aligned positions and is available through DRYAD: https://doi.org/10.5061/dryad.r7sqv9s9z. We partitioned the input alignments by locus, and for *RPB2*, by codon position. We again used RAxML for 200 likelihood searches and 500 bootstrap replicates.

## Amatoxin detection

We analyzed amatoxin concentrations from 70 specimens, from 62 *Galerina*, four *Gymnopilus*, three *Hebeloma*, and one specimen of *Flammula alnicola*. For *Galerina* specimens, we analyzed two ~5 mg replicate tissue samples for 36 of these specimens. We analyzed only one ~5 mg sample each from 26 specimens that were too small to allow replicated sampling. We tested four tissue disruption methods to compare and maximize amatoxin extraction efficiency: (1) no tissue grinding, (2) grinding with a plastic pestle, (3) grinding with a wooden stir stick and (4) vortexing the tissue with a glass bead. Tissue grinding with a wooden stir stick was most efficient and we used it for all subsequent samples. After grinding, we added 50% methanol to each tube at a ratio of 40 µL/mg starting tissue.

After 24 hours, we centrifuged samples at 13,300 rpm for 10 minutes in an accuSpin Micro 17 centrifuge (Thermo Fisher Scientific: MA, USA) and transferred the supernatant to a new 1.5 mL tube. To remove ≥ 50% of the 50% methanol solution, we spun samples for 30–60 minutes in a Savant SPD111V SpeedVac (Thermo Fisher Scientific: MA, USA) and then added sterile water to reconstitute the solution to a final volume of 200 µL. We centrifuged samples again at 13,300 rpm, for 10 minutes. Finally, we loaded 110 µL of the supernatant into individual 1.5 mL glass autosampler vials with 0.15 mL glass inserts. As a positive control, we included one vial containing 110 µL of 0.2 µg/µL α-amanitin standard (SIGMA A2263) dissolved in water. Injection volume for high-performance liquid chromatography/mass spectrometry (HPLC/MS) analysis was 100 µL.

We performed chromatographic separation using a Proto 300 C18 column (RS-2546-W185, Higgins Analytical: CA, USA) attached to an Agilent 1200 series HPLC, multi-wavelength detector, and Agilent 6120 Quadrupole MS (Agilent Technologies: CA, USA), with detection at 220, 280, 295 and 310 nm [30]. Elution solution A was 20 mM ammonium acetate pH 5 and solution B was 100% acetonitrile. The flow rate was 1 mL/min, with a gradient of 100% solution A to 100% solution B over 20 minutes. A column re-equilibration period of 10 minutes at 100% solution A was included at the end of each run.

We first determined presence or absence of α-amanitin via HPLC and UV absorbance and confirmed the results by MS. The α-amanitin standard showed an absorption peak at 310 nm at 8.5-minute retention time, coupled with strong MS signals for an ion with a mass/charge (m/z) ratio of 919. We first checked the chromatograms for each *Galerina* sample for 310 nm peaks at 8.5 minutes and we scanned extracted ion chromatogram MS data for compounds

with a mass/charge ratio of 919 at 8.5 minutes. Where UV absorbance, retention time, and MS showed evidence of α-amanitin, samples were recorded as positive. Samples were recorded as positive for β-amanitin based on a peak with the retention time of 8.0 minutes that is expected under the chromatography conditions used [30]. Samples that did not produce a distinct peak at 310 nm at 8.0 or 8.5 minutes and that lacked compounds with the expected mass/charge ratio were considered toxin-negative.

### Species delimitation

To delimit putative *Galerina* species, we used the online version of Automatic Barcode Gap Discovery (ABGD) [31] under the assumption that within species sequence variation is usually lower than the variation between species. We included the 314 ITS sequences from *Galerina*, *Psilocybe*, and *Gymnopilus* samples that were at least 500 bp long, repeating the analysis with and without a Kimura 2-parameter correction for multiple substitutions.

The ABGD software gives a range of broader or narrower estimates of species boundaries. To choose among alternative estimates, we assumed that characters of sister species evolve to show reciprocal monophyly [32], that conspecific isolates would in many cases form well-supported clades, but would lack well supported subclades [33], and that closely related species might differ in ecology [18, 34]. We did not apply a correction for multiple hits in the final analysis because preliminary results showed that a Kimura correction increased both the number of single-sequence species and the number of paraphyletic species (with no evidence of reciprocal monophyly). Our final ABGD analysis produced seven alternative estimates of possible species boundaries, based on a set of priors for the maximum percent within-species divergence that ranged from 0.001 to 0.0215. These priors bracket the range of reasonable levels of within-species divergence. The prior of 0.001 gave 71 putative species, many represented by only a single sequence and nested within another species. The prior of 0.0215 put all collections in one species in spite of many supported subclades. A prior of 0.0028 with recursive partitions resulted in 63 putative *Galerina* species, six of them nested among *G. marginata* s.l. No arbitrary prior is likely to be perfect and in some cases, the 63-group partition lumped well-supported sister taxa with consistent identifications or created paraphyletic putative species. Of the alternatives, the partition giving 63 *Galerina* species had the advantages of producing a high proportion of putative species that formed clades with moderate to high bootstrap support of 70% or more, and relatively few paraphyletic species, while dividing the *G. marginata* s. l. clade into species consistent with patterns of sequence variation in the ITS regions.

For additional support for species delimitation, we examined alignments for patterns of polymorphisms among ITS sequences from closely related putative species [34] in the *G. marginata* s.l. complex. Where collection localities of delimited species were near one another, as for many of the B.C. collections, interbreeding between close relatives with different ITS sequence variants would be expected to lead to double peaks in ITS sequences that represent heterozygosity. We examined chromatograms, correcting sequences to note double peaks in areas of otherwise clean sequence, with special attention to sites that were polymorphic across species. We considered that fixed sequence differences between sympatric populations of 10 or more specimens pointed to reproductive isolation.

## Results

### Amatoxins in the *Galerina marginata* species complex

We examined the distribution of amatoxins across the *Galerina* phylogeny (Figs 1 and 2). Of the 62 *Galerina* samples assayed, all 24 amatoxin-positive samples belonged to *G. marginata* s. l. in *Naucoriopsis* (Figs 1 and 2). We detected amatoxins in dried herbarium samples collected

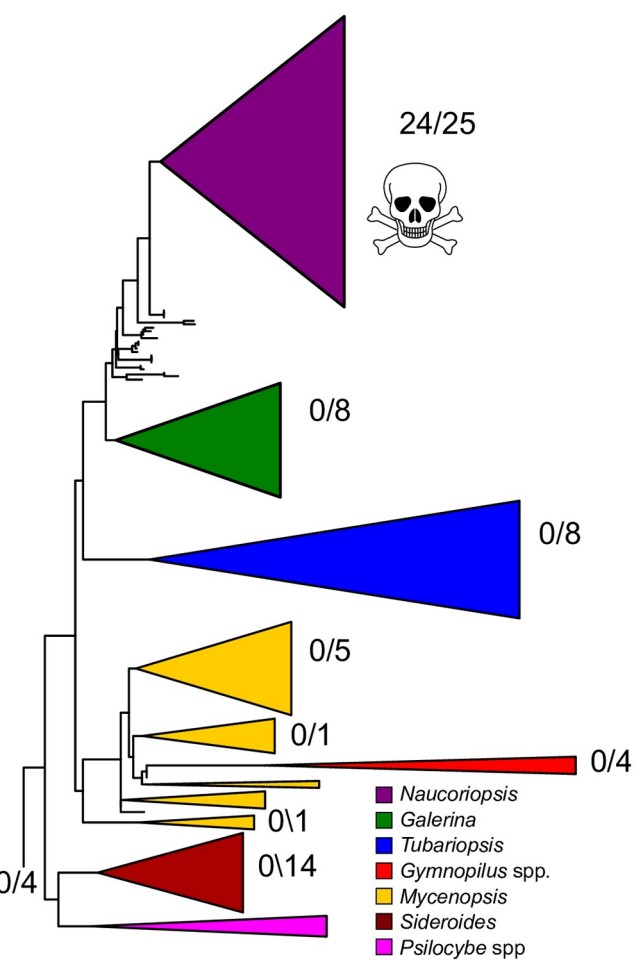

**Fig 1. All 24 toxin-positive mushroom specimens are in subgenus *Naucoriopsis* of *Galerina*.** We assayed for toxins in 70 collections representing 17 species of *Galerina* and 8 species in related genera. Each fraction is the number of samples positive for α-amanitin over the total number of specimens tested. Clade colors correspond to *Galerina* subgenera or to species of *Gymnopilus* and *Psilocybe* that appear nested in *Galerina* (S3 Table).

from 2004–2013 (S2 Table). Quantification was more difficult in samples from some of the herbarium specimens than others due to high background noise in the chromatograms. When amatoxin was detected, its concentration showed no obvious correlation with sample age (S2 Table).

The 24 samples that were positive for α-amanitin fell into two delimited species within *G. marginata* s.l.: *G. venenata* and *G. castaneipes*. Average amatoxin concentrations in *G. venenata* were significantly higher than the toxin concentration in *G. castaneipes* at P < 0.05 (t-value = 2.56; p-value = 0.018; Cohen's d = 1.1). The average toxin concentration from the nine *G. venenata* samples was 1.58 mg/g dry weight or (assuming that 88% of fresh samples was water, p. 75 in Walton [35]), ~189 μg/g estimated wet weight (S2 Table). Based on expected HPLC retention times, all nine *G. venenata* samples also contained β-amanitin. The average toxin concentration from 14 *G. castaneipes* samples was 0.99 mg/g dry weight or (assuming 88% of fresh weight is water) ~117 μg/g estimated wet weight. A peak with the expected retention time for amatoxin appeared to be present but could not be quantified in one of the 15 samples of *G. castaneipes*, and for two additional *G. castaneipes* samples, toxin concentrations were too low to quantify in at least one of the replicated measurements. Nine of the 14 *G.*

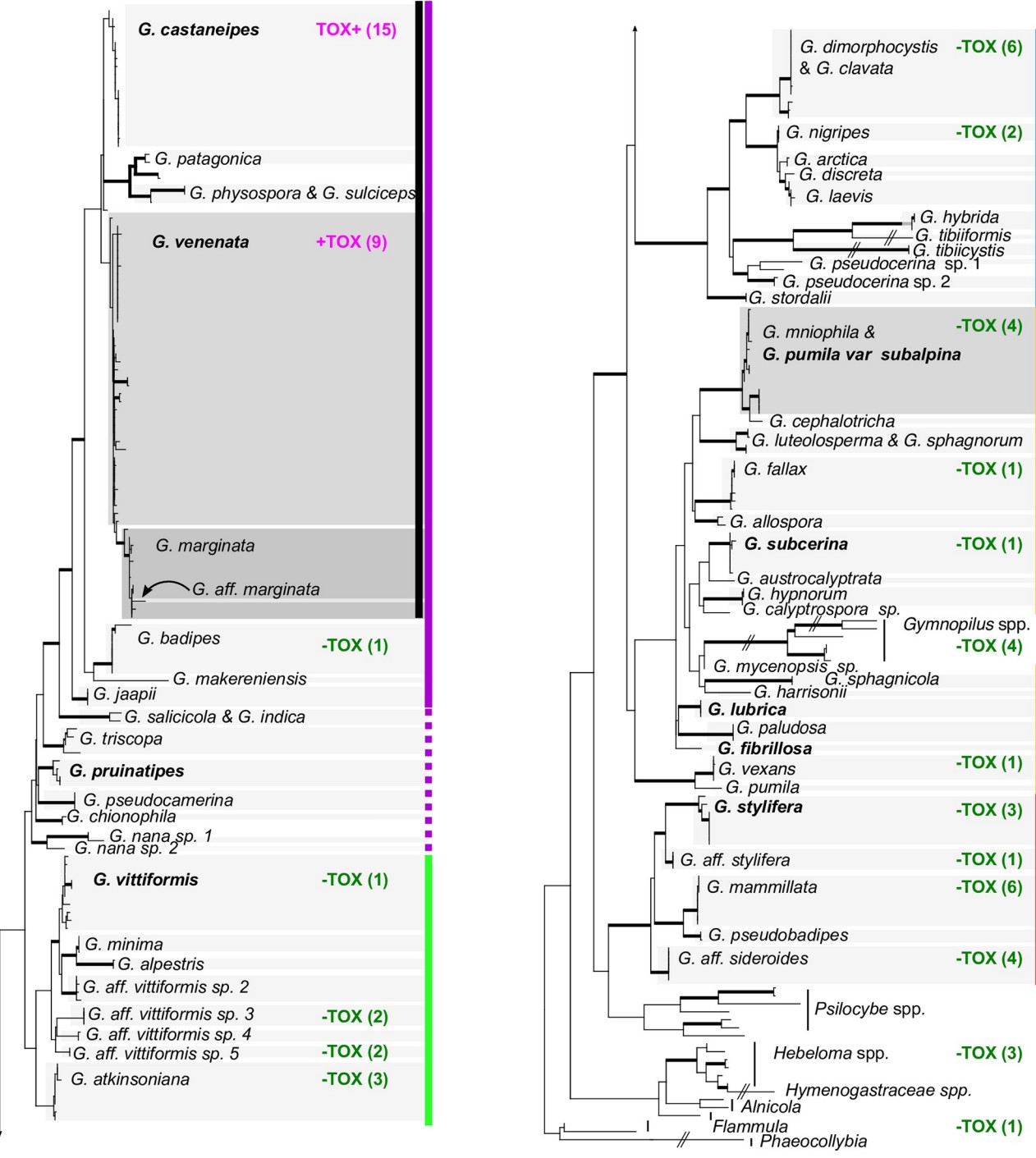

**Fig 2. Distribution of toxins across 56 species of *Galerina* and allies.** In this maximum likelihood tree, thickened branches represent 70% or more bootstrap support from concatenated ITS, LSU and *RPB2* data. Light grey boxes show monophyletic, delimited *Galerina* species. Dark grey boxes show paraphyletic species. Names outside of boxes correspond to sequences that were <500 bp long and not included in delimitations. +TOX in magenta, α-amanitin is present; -TOX in green, no amatoxins were detected; the number of collections tested is in parentheses. Vertical lines designate infrageneric groups as follows: black, *G. marginata* s.l.; solid purple, *Naucoriopsis*; dashed purple, possible *Naucoriopsis*; green, *Galerina*; blue *Tubariopsis*; gold *Mycenopsis*; red Sideroides.

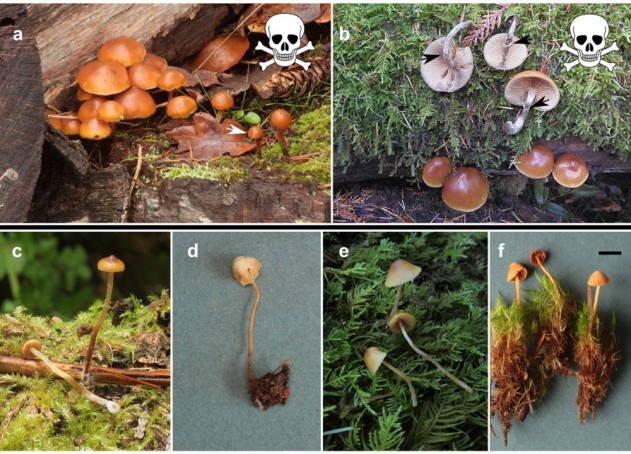

**Fig 3. Toxin containing specimens in *Galerina* subgenus *Naucoriopsis* are shown in the top row; in the lower row are examples of species in the non-toxin producing subgenera.** Each species name is followed by the specimen's UBC voucher accession number; the Mushroom Observer photograph accession number; and in italics, the name of the subgenus that includes the species. (a, b) Specimens producing positive tests for amatoxins. (a) *G. castaneipes* F28078 MO119849, *Naucoriopsis*. White arrow points to inrolled cap margin in a young mushroom. (b) *G. venenata* F26281 MO153552 *Naucoriopsis*. Black arrows point to membranous rings around the stems. (c) *G. nana* F25541 MO102538 *Naucoriopsis* (affiliation is uncertain). (d) *G. atkinsoniana* F28226 MO137762 *Galerina*. (e) *G. dimorphocystis* F25868 MO129940 *Tubariopsis*. (f) *G. subcerina* F25303 MO84732 *Mycenopsis*. (d, e) Specimens not tested, but ITS sequences match specimens without detectable toxins. (f) Specimens tested, no toxins detected. Scale bar (f) is 1 cm. Scales are not available for the other images, but estimating from the mosses and cone, caps on mushrooms (a, b) are up to ~3 cm wide. Caps on mushrooms (c-f) are ~1 cm or less wide.

*castaneipes* samples contained β-amanitin. In two samples, the presence of a β-amanitin peak was ambiguous. Four samples of *G. castaneipes* showed no trace of β-amanitin.

Amatoxins were not found in any of the genera closely related to *Galerina*; amatoxins were not detected from the four *Gymnopilus* spp., the three samples of *Hebeloma* or the sample of *Flammula alnicola*. (Figs 1 and 2). We did not detect α- or β-amanitin in *Galerina badipes* F27620, which represents the sister clade to *G. marginata* complex within *Naucoriopsis* (Fig 2 and S1 Fig). No amatoxins were detected among 37 *Galerina* samples representing the diversity of sections outside of *Naucoriopsis* (Fig 3).

## Molecular and morphological identification of toxic *Galerina*

Herbarium specimens were accurately identified to *Galerina* and its infrageneric groups (S3 Table), based on morphological identifications later confirmed by DNA barcoding. Importantly, collections of the toxin-containing *G. marginata* s.l. were usually correctly identified to this clade, and all those tested had been recognized as members of *Naucoriopsis*. This is encouraging evidence that toxic galerinas can be distinguished from other mushrooms in cases of accidental ingestion and possible poisoning, albeit with some level of expertise and with the use of microscopic characters.

Phylogenies show that many putative *Galerina* species recognized by ABGD are monophyletic, many with >70% bootstrap support (Fig 2, S1–S3 Figs; S1 and S3 Tables). However, within each infrageneric group, the application of names to species-level clades is inconsistent (S1 Fig). The inconsistency of species-level identifications even by specialists in the genus points to the lack of congruence between morphological characters and genetically defined species.

If defined phylogenetically as the sister clade to *G. badipes* (Fig 2, S1 Fig), *Galerina marginata* s.l. receives 92% bootstrap support and encompasses six putative species represented by sequences of 500 bp or longer (Fig 2, S1 Fig). Internal bootstrap support values >70% indicates that *G. marginata* s.l. has more genetic structure than expected from a single species but the putative species do not show the reciprocal monophyly expected of well-established species (S1A Fig). Collections with identical or nearly identical ITS (S2 Fig) or *RPB2* (S3 Fig) sequences were identified under various names, frequently as *G. marginata* but also as *G. autumnalis*, *G. castaneipes*, *G. oregonensis*, *G. pseudomycenopsis*, *G. unicolor* and *G. venenata* (S1 Table).

Of the toxin-containing species, *Galerina castaneipes* (Figs 3a, 4a and 4b), as delimited by ABGD, appears monophyletic in all analyses (S1–S4 Figs). It includes the type specimen *G. castaneipes* AH Smith 55523, collected on rotting oak wood in Grant's Pass, Oregon. Although conifer wood is more common in the region, all of the other 20 collections of *G. castaneipes* identified from sequencing come from collections (where wood type was recorded) from rotting hardwood, from *Quercus garryana* or *Arbutus menziesii*, geographically from the southeastern tip of Vancouver Island, British Columbia.

*Galerina venenata* contains A.H. Smith's 1953 type specimen of that species and is common among North American and European collections (Figs 2, 3b, 4c and 4d, S1 Fig). A.H. Smith's 1958 type of *G. cinnamomea* var. *cinnamomea* falls within the same clade. The *G. venenata* clade appears monophyletic in the *RPB2* tree (S3 Fig) but not in the ITS or concatenated trees with better taxon sampling (Fig 2 and S1 Fig). Collection localities of the UBC specimens of *G. venenata* and *G. castaneipes* overlapped, suggesting that parental mycelia of the two species would have had opportunities to interbreed. However, the alignment of the ITS regions shows three sites with fixed differences between the two species and little evidence of continuing genetic exchange in the form of shared ITS polymorphisms (S5 Fig). Three sequences from collections identified as species from outside *G. marginata* s.l. appeared in the *G. venenata* clade. Of these, UBC F27894 and UBC F22840 were initially identified as *G. badipes*, and UBC F24580 was identified as *G. jaapii*. On reexamination, all three specimens had predominantly 4-spored basidia, characteristic of *G. venenata*, rather than the 2-spored basidia characteristic of *G. badipes* and *G. jaapii*. The specimen UBC F24580 had a few pleurocystidia; this character and the shape of its cystidia led to its re-identification as *G. venenata*.

We label one clade "*G. marginata*" in the absence of another name that would apply to the group. No type specimen of *G. marginata* is available to clarify the application of the name. The clade receives 87% bootstrap support but to be monophyletic, it would have to include specimen *G. marginata* UWODD6MO221929, designated by ABGD as a different species (S1A Fig). Specimens identified as *G. marginata* appear in four of the putative species of *G. marginata* s.l.

**Pattern of confused application of names to species in non-toxic clades.** Application of species names is similarly problematical in subgenera *Galerina* and *Sideroides*, two clades receiving >90% bootstrap support in analyses of concatenated data (Fig 2, S1 Fig). In both of these clades, the number of monophyletic putative species is greater than the number of species names applied to collections, and application of species identifications appears almost to be random within delimited species (S1 Fig). In subgenus *Galerina*, four names are applied to collections, but eight putative species are delimited by ABGD (S3 Table). Other than the *G. alpestris* clade, each delimited species includes specimens with two or more different herbarium identifications. The clade we label as '*G. vittiformis*' includes a paratype of Smith's *G. vittiformis* var. *bispora* and specimens from N. America, Norway and Greenland. It is unclear whether this clade would also include the European type of *G. vittiformis*. Four clades labeled here as '*Galerina* aff. *vittiformis* sp. 2–5', received over 90% bootstrap support each. Some

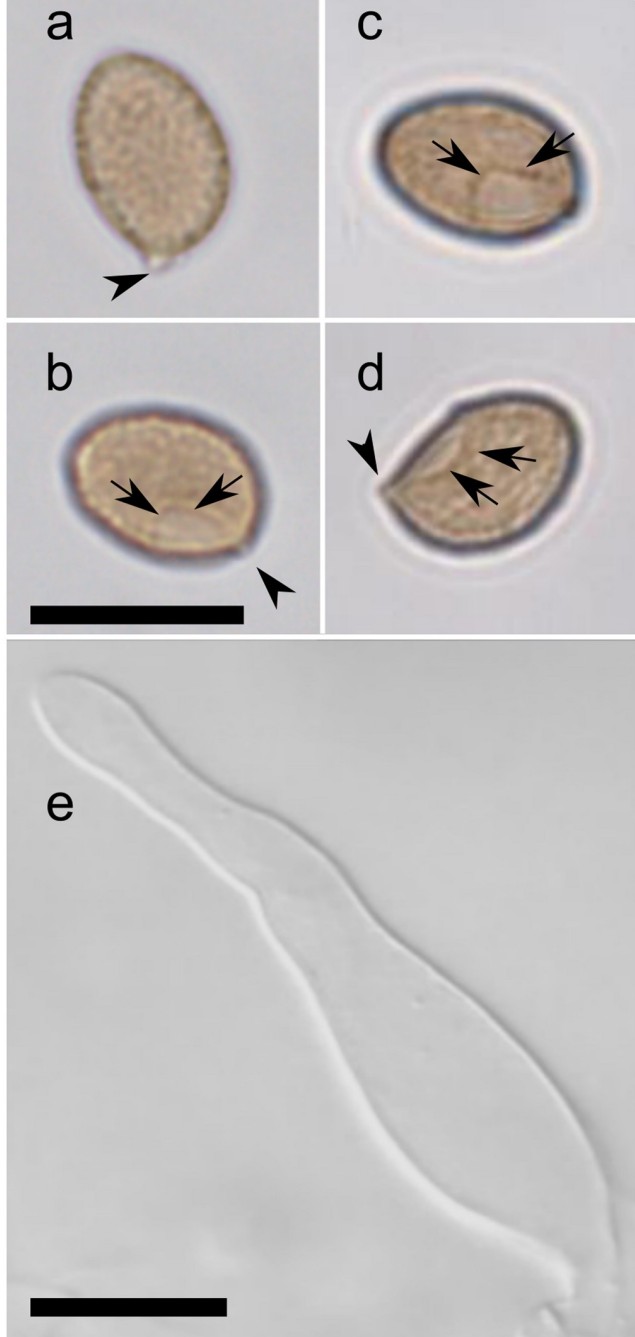

**Fig 4. Microscopic characters of toxic *Galerina marginata* complex include brown, minutely roughened spores with a plage and bottle shaped cystidia.** Although not specific to toxic *Galerina* species, these characters in any ingested mushrooms justify medical action to mitigate possible poisoning by amatoxins. (a-d) Basidiospores. (a, b) *G. castaneipes* F26244. (c-e) *G. venenata*, (c) F26281, (d) F18374, (e) cystidium of F26281. The alphanumeric codes are each specimen's UBC voucher accession number. Arrows designate the plage, the smooth area on the adaxial side of the spore just above the apiculus (arrowheads). Scale bars, 10 μm. Spores are all to the same scale.

clades appear to show geographical structuring. Sister to the *G. vittiformis* clade are five collections in two subclades, two of *G. alpestris* from Italy and in another subclade, three 'G. minima' collections from Norway and Greenland. All 13 collections of *G.* aff. *vittiformis* sp. 3 (Fig 3d) are from British Columbia; both collections of its sister taxon *G.* aff. *vittiformis* sp. 4 are from Greenland (S1 Fig). Similar problems plague naming in other clades (S3 Table).

*Galerina* **infrageneric clades.** Several *Galerina* infrageneric clades, variously considered as subgenera, sections, or stirpes in previous publications (see S3 Table and Gulden et al. [13]) receive strong support from concatenated data. The divergence order of taxa at the base of *Naucoriopsis* is unsupported but a core clade in *Naucoriopsis* that includes *G. jaapii* and *G. castaneipes* receives 79% bootstrap support (S1 Fig). *Galerina* sect. *Galerina* appears as the sister group to *Naucoriopsis*, with 95% bootstrap support from *RPB2* (S3 Fig) and 76% support from the concatenated dataset (S1 Fig). Section *Tubariopsis* appears as sister to the clade comprising *Naucoriopsis* and *Galerina*, although with <50% bootstrap support (S1 Fig).

*Gymnopilus* species are consistently nested within *Galerina* subgenus *Mycenopsis* in each individual gene tree (S2–S4 Figs) and the concatenated tree (Fig 2 and S1 Fig). A subset of species of *Mycenopsis* share a most recent common ancestor with *Gymnopilus* with 88% bootstrap support and the clade including all *Mycenopsis* and *Gymnopilus* species receives 66% bootstrap support (S1 Fig). The clade of five *Galerina* species from *Sideroides* receives 98% support from concatenated data but it is distantly related to the other *Galerina* species and instead appears, without strong support, as sister to *Psilocybe* (Fig 2, S1 Fig).

The phylogeny of *RPB2* sequences (S3 Fig) shows greater resolution and overall higher support levels for relationships among *Galerina* species compared with the phylogenies from the LSU (S4 Fig). With very low support values, the LSU phylogeny shows *Galerina* as highly paraphyletic with other genera including *Agrocybe*, *Hebeloma*, *Psilocybe* and *Cortinarius*.

## Discussion

### Toxin-producing *Galerina* species are in sect. *Naucoriopsis*

All known producers of amatoxins in *Galerina* fall into subgenus *Naucoriopsis* and most are in *Galerina marginata* s.l. This includes the 24 specimens that we identified by sequence data as *G. castaneipes* and *G. venenata*, all containing detectable amatoxin quantities. Accurate quantification from the dried specimens was difficult in some cases due to unidentifiable background peaks in chromatograms, possibly attributable to products of tissue breakdown before drying was complete. The estimated concentrations of amatoxin in fresh samples, 189 μg/g in *G. venenata* and 116 μg/g in *G. castaneipes* are comparable to 78–244 μg/g fresh weight, levels Enjalbert et al. [1] reported from 27 samples from specimens in the *G. marginata* complex. It is also comparable to amatoxin concentrations ranging from 172–367 μg/g fresh weight in *Amanita phalloides* [1].

Also in *G. marginata* s.l., in *Naucoriopsis*, and reported as toxin-positive [36], *Galerina sulciceps* is a tropical species found in greenhouses. Toxin tests and DNA sequence barcodes are not yet available for the same collection of *G. sulciceps*. The ABGD delimitation shows that the sequence from a single collection of the species is distinctive enough to be delimited along with *G. physospora* in a species separate from *G. marginata*, *G. castaneipes* and *G. venenata*. Because *G. physospora* is close to, if not synonymous with *G. sulciceps*, it seems likely to also contain amatoxins, as does *G. patagonica*, also in the *G. marginata* s.l. species complex, based on similar reasoning.

Three other species reported in the literature as toxin-positive, *Galerina beinrothii* [16], *G. helvoliceps* and *G. fasciculata* [14, 15], could not be included in our molecular analyses due to lack of DNA sequence data. *Galerina beinrothii* [37] and *G. fasciculata* [38] were originally

described as close to *G. marginata*. Smith and Singer [12] similarly placed *G. helvoliceps* near *G. marginata*. These results further support our conclusion that the amatoxin-producing *Galerina* species are found within the G. *marginata* s.l. species complex in subgenus *Naucoriopsis*.

While we detected α-amanitin in all samples tested from *G. marginata* s. l., β-amanitin was consistently present in the nine *G. venenata* samples but was undetectable from four of 15 *G. castaneipes* (S2 Table). Tyler and Smith [10] detected β-amanitin in North American samples in the initial discovery of amatoxins in *Galerina*. Besl et al. [16] detected β-amanitin in all samples assayed that contained α-amanitin. However, Luo et al. [17] did not detect β-amanitin or a gene encoding it in the published genome of *G. marginata* CBS 339.88 [39], which based on its ITS sequence (GenBank MH862132.1) falls in *G. venenata*. The β-amanitin toxin appears to be genetically encoded in *Amanita* [40, 41]. Sgambelluri et al. [30] speculated that some toxin producing fungi contain an enzyme such as a deaminase that could convert the asparagine in α-amanitin to the aspartic acid in β-amanitin. Walton (p. 75) [35] suggested that the low levels of β-amanitin peaks may also be an artifactual deamination product of α-amanitin breakdown but that the levels of β-amanitin reported by Enjalbert et al. [1] are much too high to be explained by this phenomenon.

**Toxin status in *G. badipes* (sect. *Naucoriopsis*) is uncertain.** *Galerina badipes* is the only *Galerina* species outside of *G. marginata* s.l. that is reported to contain amatoxins but we did not detect α- or β-amanitin in our sample of *G. badipes*. Besl et al. [16] detected γ-amanitin, a post-translational variant of α-amanitin [35]. Post-translational conversion of α-amanitin to γ-amanitin could explain why neither α- nor β-amanitin have been detected in *G. badipes* mushrooms, even though Luo et al. [17] detected the genes necessary for α-amanitin synthesis in a mycelial culture of the species. Further, RNA blotting showed a much weaker α-amanitin signal from *G. badipes* compared with *G. marginata* [17]. We note, however, that Luo et al. did not test for amatoxin presence using HPLC/-MS. A possible explanation that is consistent with our results and those of Besl et al. [16] is that in *G. badipes*, α-amanitin may be present but below detection limits. We believe that the UBC F27620 collection of *G. badipes* is correctly identified because its sequence matches others from *G. badipes* from Gulden et al.'s [13] study. Toxins in vouchers of *G. badipes* from across its geographical range should be analyzed. Given the confusing results, *G. badipes* has to be presumed to be toxic when implicated in accidental ingestions.

We did not test other members of sect. *Naucoriopsis* such as *G. jaapii*, which may be restricted to Europe, or other species such as *Galerina triscopa* that appear to be related to sect. *Naucoriopsis*, although with bootstrap support <50%. While this study adds to the evidence that amatoxins evolved once in the common ancestor of the *G. marginata* species complex, further analysis of additional early diverging *Naucoriopsis* species could point to earlier origin or to a more complex pattern of toxin gain and loss.

**Potential pharmaceuticals from amatoxins and associated genes from *Galerina* species.** Although best known as toxins, amatoxins and other cycloamanides may also have uses in medical therapies. Amatoxins conjugated with anti-tumor antibodies show potential for treating cancer [35, 42–44]. Cyclic peptides with other biological activities may find other uses as pharmaceutical products. Some have desirable pharmaceutical properties such as stability and rapid absorption into the bloodstream [45].

Amatoxins are expensive because they are purified from the mycorrhizal *Amanita phalloides* mushrooms [45]. Unlike the as-yet-uncultured *A. phalloides*, the saprotrophic *Galerina* species like members of *Naucoriopsis* grow at least slowly in culture, yielding from 0.5–1 mg amatoxin/g dry weight [17]. Isolating a wider range of *Galerina* species in pure culture may lead to the discovery of strains that grow faster and produce more amatoxin. Genetic

**Table 1. Comparison of characters for recognizing toxin-containing species [12, 46].**

| | Toxic | Toxins not detected |
|---|---|---|
| Species | *Galerina marginata* s.l., *G. venenata*, *G. castaneipes*, *G. badipes* | 14 species representing *Galerina* subgenera listed below. |
| Subgenera | *Naucoriopsis* | *Galerina*, *Tubariopsis*, *Mycenopsis*, *Sideroides* |
| Cap | 5–40 mm, robust compared with other *Galerina* spp.; hemispherical to convex, margin inrolled when young. | Most are < 20 mm diam.; larger in a few of the species. Delicate, conical to bell-shaped, becoming convex with age. Margin is not usually inrolled when young. |
| Stem width | 1–4 mm, commonly with membranous ring or ring zone. A ring is usually lacking in the less common species *G. castaneipes*. | Varies, but 1–2 mm in many species, mostly without a ring but white veil often present |
| Cystidia | On sides and edges of gills, ~30–70 μm long, often lageniform, rounded at base, tapering to tip or occasionally subcapitate, slightly expanded at tip. | Various; can be similar to *G. marginata* s.l., in others with more or less inflated tip; or 'tibiiform', bone-shaped with a thin, well delimited neck between an expanded base and tip. In some species only at gill edges, not on gill faces. |
| Basidiospores | Almond shaped, roughened, with a distinct plage. Spores brown, dextrinoid, turning reddish in Melzer's iodine solution. | Various, some as in *G. marginata* s.l.; others differ in shape, ornamentation, or by being completely smooth or lacking a dextrinoid reaction. |
| Habitat | On rotting wood, turf, grass or moss. | Often in moss, some on rotten wood and herbs. |

engineering may expand the range of useful cycloamanides produced from *Galerina* species' genes. Sgambelluri et al. [45] expressed *POPB*, encoding the enzyme prolyl oligopeptidase B, important in post-translational processing of amatoxins [17] from *G. marginata* in *Saccharomyces cerevisiae* to catalyze the cyclization of 100 different straight-chain peptide substrates ranging from 7–16 residues to cycloamanide configurations. The *POPB* genes from other *Galerina* species may further expand the range of potentially therapeutic cycloamanides.

## Morphological and ecological characteristics to recognize toxin producing *Galerina* in poisoning cases

Mushroom poisoning by amatoxins is difficult to diagnose because it takes two to four days after ingestion before serious symptoms appear. Basidiospores and cystidia can survive cooking or ingestion and should be sought in stomach contents or the remains of a meal containing the mushroom if poisoning is a possibility. Individual mushrooms may be atypical of their genus or species; different species often grow in close proximity and a patient may have eaten a mixture of different mushroom species. Despite these caveats, a combination of habitat, mushroom size and habit, and microscopic characters allow for recognition of *Galerina* and of the toxic species in sect. *Naucoriopsis* [46] (Table 1, Fig 4).

## Evolutionary relationships of clades within *Galerina*

The *RPB2* data contributed here improves the resolution of infrageneric relationships among *Galerina*. In contrast to phylogenies in Gulden et al. [13], our gene trees from concatenated data support *Galerina* sections *Naucoriopsis* and *Galerina* as sister clades, consistent with their shared microscopic features [11]. Also consistent with morphology, trees that include new *RPB2* sequences remove various other genera (*Phaeocollybia*, *Agrocybe*, *Alnicola*, *Hebeloma*, *Flammula*) from the nested positions within *Galerina* that they take in LSU gene trees in S4 Fig and in Gulden et al. [13].

On the other hand, this study, like Gulden et al. [13] shows *Gymnopilus* spp. evolving from within *Galerina* subgenus *Mycenopsis*. Gulden et al.'s analysis supported this relationship with a posterior probability of 1.0 from LSU data. With a smaller sample of *Gymnopilus* and *Galerina* but with *RPB2* as well as rDNA data, Matheny et al. [19] showed the same nested relationship. *Gymnopilus* and *Galerina* share spore characters including shape, ornamentation, presence of a plage and a dextrinoid reaction, and their cystidia may be similar in form, providing support for a recent shared ancestry [13, 47]. Still problematical and in need of analysis

from more loci is the unsupported sister relationship between a clade of *Psilocybe* species and five *Galerina* species that form subgenus *Sideroides*.

## Conclusion

This study combines a multi-locus sequence phylogeny with HPLC/MS toxin analysis data. The identifications of herbarium specimens to species correlated poorly with genetic species in this study as in previous analyses [13, 18], possibly because keys based on morphology fail to capture the amount of within- and among-species morphological variation. In spite of this, at a higher taxonomic level specimens are reliably identified as members of *Naucoriopsis*, the clade of species that produce toxins. Prompt morphological identification should enable recognition of likely amatoxin-containing mushrooms, speeding diagnosis and treatment for patients who have ingested these deadly toxic mushrooms.

## Supporting information

**S1 Fig. Phylogeny showing *Galerina* collections tested for amatoxins with species delimitations and country of provenance.** In this maximum likelihood tree numbers at nodes represent bootstrap support >50% from concatenated ITS, LSU and *RPB2* data. Support values are omitted from some deeply nested clades due to graphic constraints. Light grey boxes show monophyletic, delimited *Galerina* species. Darker grey boxes show delimited but paraphyletic species. A species/clade name is given in each box. Sequence names from original identifications are followed by a voucher identifier. Where applicable, the number of collections from the same country with the same sequence is given in parentheses. +TOX in magenta, α-amanitin is present; -TOX in green, no amanitins were detected. Vertical lines designate subgenera as follows: Black, *G. marginata* s. l.; solid purple, *Naucoriopsis*; dashed purple, possible *Naucoriopsis*; green, *Galerina*; blue *Tubariopsis*; gold *Mycenopsis*; red Sideroides. Orange designates *Gymnopilus* spp. nested within *Galerina*.
(DOCX)

**S2 Fig. Phylogeny of ITS sequences.** In this maximum likelihood tree of 368 sequences, thickened branches represent bootstrap support >50% from ITS data. Branch thickening is omitted from some deeply nested clades due to graphic constraints. Light grey boxes show monophyletic delimited *Galerina* species. Darker grey boxes show delimited but paraphyletic species. Sequences that are not boxed were less than 500 bp in length and not included in ABGD species delimitation. A species/clade name is given in each box. Sequence names from original identifications are followed by a voucher identifier and preceded by a number to help locate the same voucher in *RPB2* and LSU gene trees. Vertical lines designate subgenera as follows: Black, *G. marginata* s. l.; solid purple, *Naucoriopsis*; dashed purple, possible *Naucoriopsis*; green, *Galerina*; blue *Tubariopsis*; gold *Mycenopsis*; red Sideroides. Orange designates *Gymnopilus* spp. nested within *Galerina*.
(DOCX)

**S3 Fig. Phylogeny of *RPB2* sequences.** In this maximum likelihood tree with 78 taxa, numbers at nodes represent bootstrap support >70% from *RPB2* data. Support values are omitted from some deeply nested clades due to graphic constraints. Light grey boxes show monophyletic, delimited *Galerina* species. A species/clade name is given in each box. Sequence names from original identifications are followed by a voucher identifier and preceded by a number to help locate the same voucher in ITS and LSU gene trees. Vertical lines designate subgenera as follows: Solid purple, *Naucoriopsis*; dashed purple, possible *Naucoriopsis*; green, *Galerina*; blue

*Tubariopsis*; gold *Mycenopsis*; red *Sideroides*. Orange designates *Gymnopilus sapineus* nested within *Galerina*.
(DOCX)

**S4 Fig. Phylogeny of LSU sequences.** In this maximum likelihood tree with 154 taxa, numbers at nodes represent bootstrap support >70% from LSU data. Support values are omitted from some deeply nested clades due to graphic constraints. Light grey boxes show monophyletic, delimited *Galerina* species. Darker grey boxes show delimited but paraphyletic species. A species/clade name is given in each box. Sequence names from original identifications are followed by a voucher identifier and preceded by a number to help locate the same voucher in ITS and LSU gene trees. Vertical lines designate subgenera as follows: Solid purple, *Naucoriopsis*; dashed purple, possible *Naucoriopsis*; green, *Galerina*; blue *Tubariopsis*; gold *Mycenopsis*; brown *Sideroides*. Orange designates *Gymnopilus* spp. nested within *Mycenopsis*.
(DOCX)

**S5 Fig. Alignment of variable sites from the ITS region supports genetic separation of three species in *Galerina marginata* s.l.** Color of names at the left designates delimited species: *G. castaneipes*, brown; *G. venenata*, blue; and *G. marginata*, red. Under a scenario of random interbreeding, frequent heterozygosity rather than private alleles would be expected. Instead, each of the three delimited species has unique nucleotide substitutions.
(DOCX)

**S1 Table. Species delimitations, specimens, and GenBank accessions. '**Species Name ABGD delimitation' is a unique name applied to a clade within *Galerina* that is delimited by ABGD software; specimens with the same ABGD group number are in the same putative species. ABGD delimitations are only available for clades within *Galerina* and for ITS sequences > 500 bp. 'Specimen ID' designates the collection; under 'Toxin?' +TOX indicates that amatoxins were detected, -TOX indicates no amatoxins detected; 'Country' is the two-digit code for country of collection; 'Section' refers to infrageneric taxon of *Galerina*; 'Taxon #' is an arbitrary specimen tracking number also used in supplementary figures; 'Original ID' is the species name originally applied to the herbarium collection or to the sequence.
(XLSX)

**S2 Table. Results of assays to detect toxins α- and β-amanitin in *Galerina* species and closely related outgroups.** Taxon # is an arbitrary specimen tracking number used in supplementary figures; Accession number specifies a specimen in UBC; Section is the suprageneric classification of each species; Species is the identification based on sequence barcode analysis.
(XLSX)

**S3 Table. *Galerina* classification, justification and notes on the application of names to subgenera and to species.**
(DOCX)

## Acknowledgments

Jonathan Walton made this work possible by welcoming Brandon Landry into his laboratory to analyze the *Galerina* amatoxins. More broadly, Jonathan's research advanced the understanding of the toxins and their biosynthesis in a range of fungus species. His death on October 18, 2018 left a hole in the hearts of his many collaborators who benefited from his help and his ongoing generosity. Adolf Ceska provided photographs of specimens, aided in collection, and provided useful comments on the text. We thank Collection Managers Patricia Rogers

(MICH) and Olivia Lee (UBC) for loans and processing of specimens. Berni van der Meer contributed to primer design and sequencing. We thank Gro Gulden, Natural History Museum, University of Oslo, and reviewers Heather Hallen-Adams, University of Nebraska, Lincoln, and Todd Osmundson, University of Wisconsin, La Crosse for helpful suggestions.

## Author Contributions

**Conceptualization:** Anna L. Bazzicalupo, Mary L. Berbee.

**Data curation:** Brandon Landry, Anna L. Bazzicalupo, Oldriska Ceska, Mary L. Berbee.

**Formal analysis:** Brandon Landry, Anna L. Bazzicalupo, Mary L. Berbee.

**Funding acquisition:** Mary L. Berbee.

**Investigation:** Brandon Landry, Mary L. Berbee.

**Methodology:** Brandon Landry, Anna L. Bazzicalupo, Mary L. Berbee.

**Project administration:** Mary L. Berbee.

**Resources:** Oldriska Ceska, Mary L. Berbee.

**Software:** Brandon Landry, Anna L. Bazzicalupo, Mary L. Berbee.

**Supervision:** Jeannette Whitton, Anna L. Bazzicalupo, Mary L. Berbee.

**Validation:** Brandon Landry, Oldriska Ceska, Mary L. Berbee.

**Visualization:** Brandon Landry, Mary L. Berbee.

**Writing – original draft:** Brandon Landry.

**Writing – review & editing:** Brandon Landry, Jeannette Whitton, Anna L. Bazzicalupo, Oldriska Ceska, Mary L. Berbee.

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
