## [Decision Letter · Decision Letter 0]

10 Aug 2020

PONE-D-20-15894

Phylogenetic analysis of the distribution of deadly amatoxins among the little brown mushrooms of the genus Galerina

PLOS ONE

Dear Dr. Berbee,

Thank you for submitting your manuscript to PLOS ONE. After careful consideration, we feel that it has merit but does not fully meet PLOS ONE’s publication criteria as it currently stands. Therefore, we invite you to submit a revised version of the manuscript that addresses the points raised by the two referees. These are only minor comments and I am fully convinced that you are able to easily address these to improve your paper. It is likely that I will make the final decision without a second round of peer review.

We look forward to receiving your revised manuscript.

Kind regards,

Stefan Lötters

Academic Editor

PLOS ONE

Journal Requirements:

2. We noted in your submission details that a portion of your manuscript may have been presented or published elsewhere.

"No; is adapted from Landry's UBC MSc thesis but is not published or pending publication."

Please clarify whether this  publication was peer-reviewed and formally published. If this work was previously peer-reviewed and published, in the cover letter please provide the reason that this work does not constitute dual publication and should be included in the current manuscript.

Reviewers' comments:

Reviewer's Responses to Questions

**Comments to the Author**

1. Is the manuscript technically sound, and do the data support the conclusions?

Reviewer #1: Yes

Reviewer #2: Yes

2. Has the statistical analysis been performed appropriately and rigorously? 

Reviewer #1: Yes

Reviewer #2: Yes

3. Have the authors made all data underlying the findings in their manuscript fully available?

Reviewer #1: Yes

Reviewer #2: Yes

4. Is the manuscript presented in an intelligible fashion and written in standard English?

Reviewer #1: Yes

Reviewer #2: Yes

5. Review Comments to the Author

Reviewer #1: The manuscript describes a phylogenetic and toxicologic analysis of mushrooms in the genus Galerina, which contains deadly poisonous species that produce amatoxins. It is the most complete analysis of Galerina phylogeny in the context of toxicity to date, complementing but considerably expanding previous work by other authors. The analyses are appropriate and follow standard practices in fungal molecular phylogeny and in amatoxin analysis from mushroom tissue; the corresponding author is well-versed in fungal phylogeny, while the first author was trained in amatoxin analysis in the premier lab for this technique. The paper contributes to our knowledge and is well worth publishing. A few minor edits and suggestions are given below.

Line 62: Suggest “Following treatment , he slowly recovered” (“After that” sounds more spontaneous than was actually the case)

Line 64: as common names are variable, suggest adding “(Armillaria species)” after “honey mushrooms”

Lines 179-181: suggest rephrasing as “…Galerina (Strophariaceae), other Strophariaceae, and members of closely related families Hymenogastraceae, Crepidotaceae…” or some other phrasing that indicates which family Galerina is in.

Line 247: “broader or narrower”

Line 280: “amatoxin-positive”

Line 660: “Hallen-Adams HE”, not “Len-Adams HEH”

Figures S1 and S2: The bright aqua text used to indicate the types is effectively illegible (as displayed on my computer). Suggest a slightly darker color.

Reviewer #2: The manuscript by Brandon Landry and colleagues uses chemical analyses (HPLC/MS), DNA barcoding, and molecular phylogenetic analyses to determine the phylogenetic placement of species in the genus Galerina that produce deadly amatoxins. The manuscript is well written overall, its methods are suitable to its objectives, and interpretation of data is logical. I am including some specific recommendations below that I would ask the authors to address, but I do not find that any major revisions are necessary. The largest suggestion that I would make of a more general nature is to make a more active contribution to clarifying the infrageneric taxonomy of the genus. The authors stated that this is the most large-scale phylogeny of the genus to–date. This therefore seems like an excellent opportunity to clarify infrageneric nomenclature. However, even within the manuscript, these concepts seem a bit fuzzy. For example, lines 86-88 refer to "Naucoriopsis", "Galerina", "Tubariopsis", and "Mycenopsis" as provisional clade names, but lines 328-330 and line 522, for example, refer to sections, suggesting that these names have a formal taxonomic designation. At the very least, the manuscript should be consistent about referring to specific subgeneric taxa either using provisional clade names or formal sections, but not switching back and forth. Even better would be to assist in formalizing the taxonomy by establishing validly published sectional names where none exist and more clearly confirming or refuting existing formal names rather than perpetuating nomenclatural confusion where clarity could be added.

Line 19 uses the term “amatoxins,” whereas line 25 uses the term “amanitins”; I recommend being consistent with terms or at least briefly describe the difference in the beginning so the reader knows why a particular term is being used at any given point.

Line 95: should citation 19 be included here as well?

Line 115: recommend replacing "narrower species that are uncorrelated with morphological characters" with "cryptic species".

line 121: recommend changing "without Bayesian support" to "without strong Bayesian support" (there must be some support, or the clades would not be resolved). This is also an issue in a number of other places in the manuscript, including lines 344, 381, 411 and perhaps elsewhere. Specify what level you are referring to, such as “support lower than 50%”, as “no support” only makes sense if the bootstrap support value is 0%.

Line 167: recommend giving this primer a name so that others using it in the future can refer to it by name rather than "the redesigned internal forward primer reported by Landry et al."

line 181, 282, and elsewhere (many places throughout manuscript): recommend replacing "S1 table" with "table S1”, “S2 table” with ‘’table S2” etc.

line 182: recommend replacing “online server with the L-INS setting server” with “online server with the L-INS setting”

line 185: recommend replacing “best models” with “best nucleotide substitution models”

Lines 186-187 and line 200: please clarify whether RPB2 sequences were trimmed at the 3’ end. If sequences include the region between the binding sites of reverse primers 7R and 7.1R, there is an intron that should be accounted for in the jModelTest and RAxML partitions.

Line 190: recommend replacing “patterns of support over the topologies” with simply “branch support”.

Lines 236-237: Please include a description of the basis for assignment of the beta-amanitin peak, given that no standard was included.

Line 246: recommend replacing “multiple hits” with “multiple substitutions”

Line 280: recommend replacing “amatoxins-positive” with “amatoxin-positive”

Lines 314-315: “from genera closely allied with Galerina, from the four from Gymnopilus sp” reads awkwardly; suggest rewording.

Line 316: expand G. badipes to Galerina badipes, since Gymnopilus was mentioned in the previous line.

Line 344: recommend replacing “S1-S3 Figs” with “Figs. S1-S3”

Lines 344-345 and line 385: “application of names is confused and inconsistent”. The word “confused” seems unnecessarily pejorative to me; consider “inconsistent” perhaps as a substitute? It would be more informative anyway to describe the problem more specifically — are these taxonomic issues (taxa were poorly delimited in the first place), or issues of application (e.g., misidentifications because the taxa are difficult to distinguish morphologically or because the specimens were incompletely examined)?

Line 359: include a comma between “Grant's Pass” and “Oregon”

Line 393: no comma necessary after “G. vittiformis”

Lines 523-524: recommend replacing “new RPB2 remove” with “new RPB2 sequences remove”

Line 530: recommend replacing “nesting” with “nested”

Line 577: G. marginata s.l. does not appear to be designated with a black line in Fig. S2 like the caption suggests.

Line 604: Two-letter country codes should be provided with the table.

Table 1: Sideroides should be capitalized.

Fig. 1. Specify that “spp.” designates genera whereas names without “spp.” designate infrageneric groups in Galerina.

Fig. 4. Image quality is fairly low. Consider using image stacking to obtain a better image.

Fig S1: Based on the dark black line, G. patagonica, G. physospora, and G. sulciceps should be treated in G. marginata s.l., but the Discussion (lines 436-442) suggests that this is not the case; this point should be made consistent one way or the other.

6. PLOS authors have the option to publish the peer review history of their article (what does this mean?). If published, this will include your full peer review and any attached files.

Reviewer #1: **Yes: **Heather E Hallen-Adams

Reviewer #2: **Yes: **Todd Osmundson

---

## [Author Response · Author response to Decision Letter 0]

2 Dec 2020

On the following pages, we provide details about our response to reviewer comments in green. We use blue font to quote revisions to our text. 

Comments to the Author  1. Is the manuscript technically sound, and do the data support the conclusions?  The manuscript must describe a technically sound piece of scientific research with data that supports the conclusions. Experiments must have been conducted rigorously, with appropriate controls, replication, and sample sizes. The conclusions must be drawn appropriately based on the data presented.

Reviewer #1: Yes

Reviewer #2: Yes

2. Has the statistical analysis been performed appropriately and rigorously?

Reviewer #1: Yes

Reviewer #2: Yes

3. Have the authors made all data underlying the findings in their manuscript fully available?  The PLOS Data policy requires authors to make all data underlying the findings described in their manuscript fully available without restriction, with rare exception (please refer to the Data Availability Statement in the manuscript PDF file). The data should be provided as part of the manuscript or its supporting information, or deposited to a public repository. For example, in addition to summary statistics, the data points behind means, medians and variance measures should be available. If there are restrictions on publicly sharing data—e.g. participant privacy or use of data from a third party—those must be specified.

Reviewer #1: Yes

Reviewer #2: Yes

4. Is the manuscript presented in an intelligible fashion and written in standard English?  PLOS ONE does not copyedit accepted manuscripts, so the language in submitted articles must be clear, correct, and unambiguous. Any typographical or grammatical errors should be corrected at revision, so please note any specific errors here.

Reviewer #1: Yes

Reviewer #2: Yes

5. Review Comments to the Author  Please use the space provided to explain your answers to the questions above. You may also include additional comments for the author, including concerns about dual publication, research ethics, or publication ethics. (Please upload your review as an attachment if it exceeds 20,000 characters)

Reviewer #1: The manuscript describes a phylogenetic and toxicologic analysis of mushrooms in the genus Galerina, which contains deadly poisonous species that produce amatoxins. It is the most complete analysis of Galerina phylogeny in the context of toxicity to date, complementing but considerably expanding previous work by other authors. The analyses are appropriate and follow standard practices in fungal molecular phylogeny and in amatoxin analysis from mushroom tissue; the corresponding author is well-versed in fungal phylogeny, while the first author was trained in amatoxin analysis in the premier lab for this technique. The paper contributes to our knowledge and is well worth publishing. A few minor edits and suggestions are given below.  Line 62: Suggest “Following treatment , he slowly recovered” (“After that” sounds more spontaneous than was actually the case)

Done.

 Line 64: as common names are variable, suggest adding “(Armillaria species)” after “honey mushrooms”

Done.

 Lines 179-181: suggest rephrasing as “…Galerina (Strophariaceae), other Strophariaceae, and members of closely related families Hymenogastraceae, Crepidotaceae…” or some other phrasing that indicates which family Galerina is in.

Done. Revised as:

Galerina, Hebeloma and Gymnopilus, genera representing the family Hymenogastraceae.

 Line 247: “broader or narrower”

Done.

 Line 280: “amatoxin-positive”

Done Line 660: “Hallen-Adams HE”, not “Len-Adams HEH”

Corrected

 Figures S1 and S2: The bright aqua text used to indicate the types is effectively illegible (as displayed on my computer). Suggest a slightly darker color.

Agreed. We now use a darker blue that shows up better.

Reviewer #2: The manuscript by Brandon Landry and colleagues uses chemical analyses (HPLC/MS), DNA barcoding, and molecular phylogenetic analyses to determine the phylogenetic placement of species in the genus Galerina that produce deadly amatoxins. The manuscript is well written overall, its methods are suitable to its objectives, and interpretation of data is logical. I am including some specific recommendations below that I would ask the authors to address, but I do not find that any major revisions are necessary. 

The largest suggestion that I would make of a more general nature is to make a more active contribution to clarifying the infrageneric taxonomy of the genus. The authors stated that this is the most large-scale phylogeny of the genus to–date. This therefore seems like an excellent opportunity to clarify infrageneric nomenclature. However, even within the manuscript, these concepts seem a bit fuzzy. For example, lines 86-88 refer to "Naucoriopsis", "Galerina", "Tubariopsis", and "Mycenopsis" as provisional clade names, but lines 328-330 and line 522, for example, refer to sections, suggesting that these names have a formal taxonomic designation. At the very least, the manuscript should be consistent about referring to specific subgeneric taxa either using provisional clade names or formal sections, but not switching back and forth. Even better would be to assist in formalizing the taxonomy by establishing validly published sectional names where none exist and more clearly confirming or refuting existing formal names rather than perpetuating nomenclatural confusion where clarity could be added.

Done. Table S3 now shows that we are formally emending infrageneric taxa that have been recognized at various ranks to subgenera. 

This was a good suggestion because it simplifies the story while imposing minimal nomenclatoral change. We base most of the subgenera on clades already recognized by Gulden & Hallgrímsson 2000. The subgenera are consistent with informal clade names in the phylogeny of the LSU region from Gulden et al. 2005.   Line 19 uses the term “amatoxins,” whereas line 25 uses the term “amanitins”; I recommend being consistent with terms or at least briefly describe the difference in the beginning so the reader knows why a particular term is being used at any given point.

Replaced 'amanitins' with 'amatoxin' when referring to general group of toxins, applying 'amanitin' when referring to a specific compound, e.g. β-amanitin.

Added explanation: Prompted by these poisoning cases, Tyler and Smith [10] used paper chromatography to show that G. venenata contains α- and β-amanitin – two of the amatoxins, the toxic peptides identified from the genus Amanita. 

 Line 95: should citation 19 be included here as well?

Not done because ref 19 is no longer appropriate due to edits that changed context.  Line 115: recommend replacing "narrower species that are uncorrelated with morphological characters" with "cryptic species".

Done. Is now shorter and more direct.  line 121: recommend changing "without Bayesian support" to "without strong Bayesian support" (there must be some support, or the clades would not be resolved). 

Done.

This is also an issue in a number of other places in the manuscript, including lines 344, 381, 411 and perhaps elsewhere. Specify what level you are referring to, such as “support lower than 50%”, as “no support” only makes sense if the bootstrap support value is 0%.

Done as specified in italics below.

Line 267: proportion of putative species that formed clades with moderate to high bootstrap support of 70% or more,  Line 346: monophyletic, many with >70% bootstrap support

Line 353: Internal bootstrap support values >70% 

Line 391: two clades receiving >90% bootstrap support 

Line 414: Naucoriopsis and Galerina, although with <50% bootstrap support 

Line 482: although with bootstrap support <50%

 Line 167: recommend giving this primer a name so that others using it in the future can refer to it by name rather than "the redesigned internal forward primer reported by Landry et al."

It is now berniF, named for Berni van der Meer who designed it.

 line 181, 282, and elsewhere (many places throughout manuscript): recommend replacing "S1 table" with "table S1”, “S2 table” with ‘’table S2” etc.

Not changed because "S1 Table" is the journal-specified format. 

line 182: recommend replacing “online server with the L-INS setting server” with “online server with the L-INS setting”

ln 183, Done  line 185: recommend replacing “best models” with “best nucleotide substitution models” 

Done. we selected as best nucleotide substitution models

 Lines 186-187 and line 200: please clarify whether RPB2 sequences were trimmed at the 3’ end. If sequences include the region between the binding sites of reverse primers 7R and 7.1R, there is an intron that should be accounted for in the jModelTest and RAxML partitions.

I double-checked and the intron had been spliced out of our alignment before analysis.

This means that the text was OK as originally submitted:

Line 184: For the RPB2 dataset, we excluded introns from the final alignment.

 Line 190: recommend replacing “patterns of support over the topologies” with simply “branch support”.

Line 191 Done. We used 500 bootstrap replicates to assess branch support.  Lines 236-237: Please include a description of the basis for assignment of the beta-amanitin peak, given that no standard was included.

We add an explanation that the basis is retention time, under the conditions used for chromatography of mushroom amanitins, and we added a citation of the paper that best documents the positions of the amanitin peaks under the conditions used in our paper.

Samples were recorded as positive for �-amanitin based on a peak with the retention time of 8.0 minutes that is expected under the chromatography conditions used [31].   Line 246: recommend replacing “multiple hits” with “multiple substitutions”

Line 269 done.   Line 280: recommend replacing “amatoxins-positive” with “amatoxin-positive” Line 307 done.

 Lines 314-315: “from genera closely allied with Galerina, from the four from Gymnopilus sp” reads awkwardly; suggest rewording.

Line 316. Reworded:

Amatoxins were not found in any of the genera closely related to Galerina; amatoxins were not detected from the four Gymnopilus spp., the three samples of Hebeloma or the sample of Flammula alnicola. (Figs 1 and 2).  Line 316: expand G. badipes to Galerina badipes, since Gymnopilus was mentioned in the previous line. Done.

 Line 344: recommend replacing “S1-S3 Figs” with “Figs. S1-S3” Not changed because "S1 Figs" is the journal-specified format.  Lines 344-345 and line 385: “application of names is confused and inconsistent”. The word “confused” seems unnecessarily pejorative to me; consider “inconsistent” perhaps as a substitute? It would be more informative anyway to describe the problem more specifically — are these taxonomic issues (taxa were poorly delimited in the first place), or issues of application (e.g., misidentifications because the taxa are difficult to distinguish morphologically or because the specimens were incompletely examined)? 

Corrected and we agree that pejorative implications are unwarranted. We don't discuss the reasons for the lack of congruence because we don't have detailed morphological notes for most specimens that would allow us to go further.

Line 381 ...application of names to species-level clades is inconsistent (S1 Fig). The inconsistency of species-level identifications even by specialists in the genus points to lack of congruence between morphological characters and genetically defined species.

 Line 359: include a comma between “Grant's Pass” and “Oregon” Done

 Line 393: no comma necessary after “G. vittiformis” Done

 Lines 523-524: recommend replacing “new RPB2 remove” with “new RPB2 sequences remove” Done Line 530: recommend replacing “nesting” with “nested” Done

Line 577: G. marginata s.l. does not appear to be designated with a black line in Fig. S2 like the caption suggests.  Line 604: Two-letter country codes should be provided with the table. S1 Table does specify the two-letter country codes in column D. Table 1: Sideroides should be capitalized.

Done Fig. 1. Specify that “spp.” designates genera whereas names without “spp.” designate infrageneric groups in Galerina.

Thanks, corrected.

Clade colors correspond to Galerina subgenera or to species of Gymnopilus and Psilocybe that appear nested in Galerina (S3 Table). 

Fig. 4. Image quality is fairly low. Consider using image stacking to obtain a better image. The images do clearly show the features that are important in recognizing toxic species, and so we would have difficulty improving them at this point due to covid-related problems with access to the herbarium/specimens and to the camera. 

Fig S1: Based on the dark black line, G. patagonica, G. physospora, and G. sulciceps should be treated in G. marginata s.l., but the Discussion (lines 436-442) suggests that this is not the case; this point should be made consistent one way or the other.

We do consider all of those species to be part of G. marginata s.l. and clarify this as follows: 

line 440 Also in G. marginata s.l., in Naucoriopsis, and also toxin-positive, Galerina sulciceps...

line 445 G. patagonica, also in the G. marginata s.l. species complex

6. PLOS authors have the option to publish the peer review history of their article (what does this mean?). If published, this will include your full peer review and any attached files.   Do you want your identity to be public for this peer review? For information about this choice, including consent withdrawal, please see our Privacy Policy.

Reviewer #1: Yes: Heather E Hallen-Adams

Reviewer #2: Yes: Todd Osmundson

 While revising your submission, please upload your figure files to the Preflight Analysis and Conversion Engine (PACE) digital diagnostic tool, https://pacev2.apexcovantage.com/. PACE helps ensure that figures meet PLOS requirements. To use PACE, you must first register as a user. Registration is free. Then, login and navigate to the UPLOAD tab, where you will find detailed instructions on how to use the tool. If you encounter any issues or have any questions when using PACE, please email PLOS at figures@plos.org. Please note that Supporting Information files do not need this step.

---

## [Decision Letter · Decision Letter 1]

20 Jan 2021

PONE-D-20-15894R1

Phylogenetic analysis of the distribution of deadly amatoxins among the little brown mushrooms of the genus Galerina

PLOS ONE

Dear Dr. Berbee,

Thank you for submitting your manuscript to PLOS ONE. After careful consideration, we feel that it has merit but does not fully meet PLOS ONE’s publication criteria as it currently stands. Therefore, we invite you to submit a revised version of the manuscript that addresses the points raised during the review process.

Both referees find the paper sound, as it is. So do I. There are a few minor aspects, raised by both referees, that need consideration by the authors before final acceptance. This is 'cosmetics', basically (see comments to authors).

We look forward to receiving your revised manuscript.

Kind regards,

Stefan Lötters

Academic Editor

PLOS ONE

Reviewers' comments:

Reviewer's Responses to Questions

**Comments to the Author**

1. If the authors have adequately addressed your comments raised in a previous round of review and you feel that this manuscript is now acceptable for publication, you may indicate that here to bypass the “Comments to the Author” section, enter your conflict of interest statement in the “Confidential to Editor” section, and submit your "Accept" recommendation.

Reviewer #1: All comments have been addressed

Reviewer #2: All comments have been addressed

2. Is the manuscript technically sound, and do the data support the conclusions?

Reviewer #1: Yes

Reviewer #2: Yes

3. Has the statistical analysis been performed appropriately and rigorously? 

Reviewer #1: Yes

Reviewer #2: Yes

4. Have the authors made all data underlying the findings in their manuscript fully available?

Reviewer #1: Yes

Reviewer #2: Yes

5. Is the manuscript presented in an intelligible fashion and written in standard English?

Reviewer #1: Yes

Reviewer #2: Yes

6. Review Comments to the Author

Reviewer #1: A tiny point: Line 98: there appears to be an extra “G.” in “G. G. beinrothii”. I will now include this sentence because it tells me I have not met the character count.

Reviewer #2: I commend Brandon Landry and colleagues on their careful revision of the manuscript. The authors addressed all comments with either corrections or reasonable justifications. The nomenclatural details that I mentioned in the previous review are handled very well in the revision. I include a few minor points below that I think could improve clarity or fix typographic errors. The manuscript provides a great contribution to our understanding of amatoxin evolution in Galerina, and improves the predictive framework for determining which other species in the genus are toxic. I look forward to seeing how these predictions perform when toxicity data are collected for additional species. In addition, the paper improves the resolution of infrageneric relationships in Galerina, and points to some areas where further phylogenetic studies are needed (such as the relationship between Galerina subgen. Mycenopsis and Gymnopilus, and the relationship between Galerina subgen. Sideroides and Psilocybe) -- it will be interesting to see how these questions are resolved with future research.

Line 44: italicize Galerina

Line 57: recommend replacing “on liver” with “on the liver”

Line 84: suggest “subgenera and sections; e.g., Gulden and Hallgrímsson [11], and Smith and Singer [12].”

Line 98: recommend replacing “G. G. beinrothii” with “G. beinrothii”

Line 161: missing period after “amplifications”

Line 424: recommend replacing “without support” to “without strong support”

Line 426: recommend replacing “relations” with “relationships”

Line 442: G. sulciceps is reported here as toxin-positive, but that result is not shown in Fig. 2 or Fig. S1 or Table S1; I assume that the information on toxicity is therefore from the literature and that the sequenced specimen is not the one that was tested. I recommend clarifying this point, along with a brief explanation of the basis for considering the reported specimen and the sequenced specimen as conspecific.

Line 449: Similarly, I recommend clarifying this line, replacing “Other toxin-positive specimens without DNA barcodes” to something like “Other species reported in the literature as toxin-positive, but not included in our molecular analyses due to lack of DNA sequence data”

Line 452: consider appending something like “These results further support our conclusion that the amatoxin-producing Galerina species are found within the G. marginata s.l. species complex in subgenus Naucoriopsis.”

Line 453: recommend replacing “from the G. marginata s. l.” with “from G. marginata s. l.”

7. PLOS authors have the option to publish the peer review history of their article (what does this mean?). If published, this will include your full peer review and any attached files.

Reviewer #1: **Yes: **Heather Hallen-Adams

Reviewer #2: **Yes: **Todd Osmundson

---

## [Author Response · Author response to Decision Letter 1]

21 Jan 2021

The reviewers pointed out typos or places where clarification was needed. None were complicated and all were helpful. We made all of the corrections as recommended.

Reviewer #1: A tiny point: Line 98: there appears to be an extra “G.” in “G. G. beinrothii”.

Done.

 I will now include this sentence because it tells me I have not met the character count.

Reviewer #2: I commend Brandon Landry and colleagues on their careful revision of the manuscript. The authors addressed all comments with either corrections or reasonable justifications. The nomenclatural details that I mentioned in the previous review are handled very well in the revision. I include a few minor points below that I think could improve clarity or fix typographic errors. The manuscript provides a great contribution to our understanding of amatoxin evolution in Galerina, and improves the predictive framework for determining which other species in the genus are toxic. I look forward to seeing how these predictions perform when toxicity data are collected for additional species. In addition, the paper improves the resolution of infrageneric relationships in Galerina, and points to some areas where further phylogenetic studies are needed (such as the relationship between Galerina subgen. Mycenopsis and Gymnopilus, and the relationship between Galerina subgen. Sideroides and Psilocybe) -- it will be interesting to see how these questions are resolved with future research.

Line 44: italicize Galerina

Done.

Line 57: recommend replacing “on liver” with “on the liver”

Done.

Line 84: suggest “subgenera and sections; e.g., Gulden and Hallgrímsson [11], and Smith and Singer [12].”

Done.

Line 98: recommend replacing “G. G. beinrothii” with “G. beinrothii”

Done.

Line 161: missing period after “amplifications”

Done.

Line 424: recommend replacing “without support” to “without strong support”

Done.

Line 426: recommend replacing “relations” with “relationships”

Done.

Line 442: G. sulciceps is reported here as toxin-positive, but that result is not shown in Fig. 2 or Fig. S1 or Table S1; I assume that the information on toxicity is therefore from the literature and that the sequenced specimen is not the one that was tested. I recommend clarifying this point, along with a brief explanation of the basis for considering the reported specimen and the sequenced specimen as conspecific.

Done.

Also in G. marginata s.l., in Naucoriopsis, and reported as toxin-positive [36], Galerina sulciceps is a tropical species found in greenhouses. Toxin tests and DNA sequence barcodes are not yet available for the same collection of G. sulciceps. 

Line 449: Similarly, I recommend clarifying this line, replacing “Other toxin-positive specimens without DNA barcodes” to something like “Other species reported in the literature as toxin-positive, but not included in our molecular analyses due to lack of DNA sequence data”

Done, edited as:

Three species reported in the literature as toxin-positive, Galerina beinrothii [16], G. helvoliceps and G. fasciculata [14, 15], could not be included in our molecular analyses due to lack of DNA sequence data.

Line 452: consider appending something like “These results further support our conclusion that the amatoxin-producing Galerina species are found within the G. marginata s.l. species complex in subgenus Naucoriopsis.”

Done.

Line 453: recommend replacing “from the G. marginata s. l.” with “from G. marginata s. l.”

Done.

---

## [Editor Report · Decision Letter 2]

22 Jan 2021

Phylogenetic analysis of the distribution of deadly amatoxins among the little brown mushrooms of the genus Galerina

PONE-D-20-15894R2

Dear Dr. Berbee,

We’re pleased to inform you that your manuscript has been judged scientifically suitable for publication and will be formally accepted for publication once it meets all outstanding technical requirements.

Kind regards,

Stefan Lötters

Academic Editor

PLOS ONE
---

## [Editor Report · Acceptance letter]

29 Jan 2021

PONE-D-20-15894R2 

Phylogenetic analysis of the distribution of deadly amatoxins among the little brown mushrooms of the genus *Galerina*

Dear Dr. Berbee:

I'm pleased to inform you that your manuscript has been deemed suitable for publication in PLOS ONE. Congratulations! Your manuscript is now with our production department. 

Kind regards, 

on behalf of

Prof. Dr. Stefan Lötters 

Academic Editor

PLOS ONE